# Cultural Identity Conflict Informs Engagement with Self-Management Behaviours for South Asian Patients Living with Type-2 Diabetes: A Critical Interpretative Synthesis of Qualitative Research Studies

**DOI:** 10.3390/ijerph18052641

**Published:** 2021-03-05

**Authors:** Tasneem Patel, Kanayo Umeh, Helen Poole, Ishfaq Vaja, Lisa Newson

**Affiliations:** 1Department of Primary Care and Mental Health, Institute of Population Health, University of Liverpool, Liverpool L69 3BX, UK; Tasneem.Patel@liverpool.ac.uk; 2Faculty of Health, School of Psychology, Liverpool John Moores University, Liverpool L3 3AF, UK; F.K.Umeh@ljmu.ac.uk (K.U.); H.M.Poole@ljmu.ac.uk (H.P.); ishfaq.vaja@bthft.nhs.uk (I.V.); 3NHS Bradford Teaching Hospital, Bradford BD9 6RJ, UK

**Keywords:** type 2 diabetes, South Asian, culture, conflict, healthcare, qualitative, evidence synthesis

## Abstract

The prevalence of type-2 diabetes (T2D) is increasing, particularly among South Asian (SA) communities. Previous research has highlighted the heterogeneous nature of SA ethnicity and the need to consider culture in SA patients’ self-management of T2D. We conducted a critical interpretative synthesis (CIS) which aimed to a) develop a new and comprehensive insight into the psychology which underpins SA patients’ T2D self-management behaviours and b) present a conceptual model to inform future T2D interventions. A systematic search of the literature retrieved 19 articles, including 536 participants. These were reviewed using established CIS procedures. Analysis identified seven constructs, from which an overarching synthesizing argument ‘Cultural Conflict’ was derived. Our findings suggest that patients reconstruct knowledge to manage their psychological, behavioural, and cultural conflicts, impacting decisional conflicts associated with T2D self-management and health professional advice (un)consciously. Those unable to resolve this conflict were more likely to default towards cultural identity, continue to align with cultural preferences rather than health professional guidance, and reduce engagement with self-management. Our synthesis and supporting model promote novel ideas for self-management of T2D care for SA patients. Specifically, health professionals should be trained and supported to explore and mitigate negative health beliefs to enable patients to manage social-cultural influences that impact their self-management behaviours.

## 1. Introduction

Over the past 60–70 years, the UK population has become more ethnically diverse. The UK’s most recent census in 2011 [1] recorded 1,451,862 residents of Indian, 1,174,983 of Pakistani, and 451,529 of Bangladeshi ethnicity, equalling a South Asian (SA) population of 4.9 per cent of the UK total (excluding other Asian groups and people of mixed ethnicity). As a country, the UK makes-up a significant portion of the SA migrant population. Although still a minority, nearly ten years later, SA ethnicity now represents a significant proportion of the UK population and is expected to increase [2,3]. It was estimated that ethnic groups could make up 30% of the UK population by 2061 [2,3]. However, although often grouped as one, SA ethnicity represents a heterogeneous group of individuals coming from and representing diverse cultures, communities, and countries [4]. For example, SA individuals may identify themselves via a religious label [5]; according to the 2011 census, the three most prominent religions in UK SAs were Hinduism, Islam and Sikhism [6]. Alternatively, they may identify themselves based on languages and dialects associated with SA sub-groups: Gujarati; Hindi; Urdu; Punjabi; Bengali; Sylheti [7]. It is also noteworthy that over the last 50 years, there have been at least two generations of SAs born and educated in the UK [8]. Thus, the cultural characteristics of UK SAs are both diverse and changing.

Type 2 diabetes (T2D) is a worldwide epidemic [9], with an estimated prevalence of 415 million adults aged between 20 and 70 years old [10]. By 2040, T2D will affect one in ten people worldwide [11]. The prevalence of T2D is increasing in the UK [12] and within the SA community [13,14,15,16,17,18]. From the age of 25, people of a SA ethnicity are significantly more likely to develop T2D than the White population [14,16,17]. SA people are more likely to experience more significant T2D complications than the UK’s general population [19,20]. Such health inequalities and health complications are even more apparent since the COVID-19 pandemic [21], which shows greater risks for Black and SA individuals [22]. Specifically, Black and SA individuals with severe COVID-19 and diabetes are at significantly increased risk of mortality [23]. Effective diabetes self-management is necessary to obtain optimum glycaemic control, reducing the risk and severity of comorbid complications, such as COVID-19.

According to NHS England [24], ‘People have a key role in protecting their own health, choosing appropriate treatments and managing long-term conditions. Self-management is a term used to include all the actions taken by people to recognise, treat and manage their health. They may do this independently or in partnership with the healthcare system’. Clinical guidelines [25] recommend self-management behaviours for T2D to ensure optimal clinical outcomes; these include attending structured education and making lifestyle changes such as dietary modifications, taking part in physical activity (PA), self-monitoring of blood glucose and taking prescribed medications (as appropriate).

Health disparities exist for SA people in the UK [26]. Previous research has explored healthcare delivery acknowledging difficulties in health literacy, language, and low attendance at health education sessions [27,28,29]. Patient-centred care should be offered to people with T2D [30]. ‘Treatment and care should take into account individual needs and preferences’ [25]. Despite this call for personalised and tailored care, previous research has consistently highlighted challenges for SA people to engage with the recommended T2D self-management behaviours [31,32,33,34].

Insights into the phenomenon of T2D self-management behaviours, specifically into SA patients as an ethnic minority group, may be developed through qualitative research, which can present a rich account of a person’s experiences [35]. These accounts offer an opportunity for theoretical development and relevance for innovation in T2D healthcare delivery. Qualitative synthesis’ combines and offers an opportunity to reanalyse findings collectively across a range of qualitative studies [36]. A synthesis helps to negate criticism that qualitative studies are limited in their transferability (e.g., small sample sizes). Qualitative syntheses have been published on evidence across several health conditions [37,38] and have been used to recommend new theoretical developments and new interventions or implementation strategies.

To date, two qualitative reviews have been published on diabetes self-management in the SA population [39,40]. Fleming and Gillibrand’s [39] review aimed to explore how SA people self-manage diabetes and identify nursing practice improvements. Published in 2009, the review included eleven articles (from nine studies published between 1992 and 2005, totalling *n* = 408 participants), of which ten articles reported on SAs living in the UK. Consideration for the overall quality of the articles included in the review was not available. The review highlighted the heterogeneous nature of SA ethnicity and the diversity in their beliefs and behaviours. The authors advised caution in considering SA people as one homogenous minority group. They suggested that SA culture may impact T2D self-management, but nursing practice should not assume that SA people hold specific beliefs. Fleming and Gillibrand recommended that care be personalised as appropriate, although the review does not provide specific guidance on considering culture within personalised care. The Sohal et al. [40] review (included mixed methods, although most studies used qualitative methodology) investigated SA patients’ views, attitudes, or beliefs on barriers and facilitators in the core components of diabetes management. The review included twenty articles (*n* = 1980 participants, of whom the majority were female). Of the twenty articles, eleven were conducted in the UK (eight in England, three in Scotland). While published in 2015, most of the included articles were published before 2010, with the first article being published in 1994. The overall quality varied: though most were low quality, with limited information on analytical strategies. Sohal et al. [40] conducted an interpretive meta-ethnographic analysis and reported that diabetes self-management could be improved by targeting culturally specific issues. Both reviews [39,40] included participants from diverse SA ethnic backgrounds (Indian Malaysian, Bangladeshi, Pakistani, White Kashmiri, Afro-Caribbean, and Indian). However, Flemming et al. [39] acknowledged that half of the articles recruited Muslim participants, so the sample was skewed towards this sub-group of SAs.

These previous reviews [39,40] provide a foundation for understanding diabetes self-management, although it is noteworthy that the evidence presented across these two reviews is dated. The UK’s SA population has changed [41,42] and is expected to increase dramatically [2,3]. There is a significant increase in T2D prevalence in the UK [12] with a high prevalence in the SA population who are four to six time more likely to develop T2D [13,15,16,17,18], and in the last ten years, significant changes have and will continue to occur within the NHS [43,44], and specifically for diabetes services [25,30]. Hence, there is a need to reconsider the evidence-base and conduct a new interpretative review to understand self-management behaviours and the nuances for SA people.

To address this need, we systematically retrieved qualitative studies of T2D self-management behaviours in SA patients and conducted a critical interpretative synthesis (CIS) [45] on this evidence. CIS enables the interrogation of data from different methodologies to produce a new analysis and synthesising argument [46] to create a new understanding of the existing literature [47]. This CIS reports a reanalysis of the qualitative evidence to (a) develop a new and comprehensive insight into the underpinning psychology, which underpins SA patients’ self-management behaviours and (b) present a conceptual model to inform future T2D self-management interventions or delivery for this patient group.

## 2. Materials and Methods

### 2.1. Study Design

To employ both a theoretical and critical evaluation, we conducted a critical interpretive synthesis (CIS) [45]. CIS is an adaptation of meta-ethnography, as well as borrowing techniques from grounded theory. The epistemological position of CIS articulates that of an interpretative stance. The process promotes the development of a new theoretical model of a phenomenon. The CIS is not just a systematic review but also creates new knowledge. The specific question for this review remained open to modification as per the CIS process. The conduct of this review and the presentation of methodology and analysis within this article has aligned to the Enhancing Transparency in Reporting the Synthesis of Qualitative Research (ENTREQ) checklist [48] (Appendix A).

### 2.2. Literature Search and Eligibility

In line with the CIS process, the search strategy was adaptive, and data extraction employed an iterative process, adopting an interpretive stance. The search sought all articles on Type 2 Diabetes, Adherence, South Asian, Lifestyle, and Qualitative (Appendix A). The term qualitative ’and’ was attached to all the search terms to filter for qualitative articles (any type of qualitative article was acceptable). The research platform EBSCOhost was used to search a range of databases (such as Cinahl; Pub Med; Psych Info; Psych Articles; Web of Knowledge/Web Of Science; Scopus; Cogprints Social Science; Open Access Repository (Appendix A). Based on an initial assessment by the first author and discussion with the last author, articles were subject to systematic screening [49] (pp. 102–108) of the titles, abstracts or full text as deemed to meet the inclusion criteria. The first search sought articles published up to July 2016. Subsequently, we employed berry-picking [50], which consisted of 6 additional search strategies (including grey literature, footnote chasing, citation searching, journal run, area scanning, subject searches in bibliographies, abstracting and indexing service, and author searching). The initial screening assessed if the article (1) described a qualitative research study and (2) focused on the target population (South Asian), and (3) explored experiences relating to T2D self-management. Eligible articles were read in full to assess eligibility and relevance to the review aims. In line with CIS methodology, theoretical saturation, and to ensure the review included all current and relevant evidence, the search process was repeated (June 2020) to include articles published between July 2016 and May 2020.

### 2.3. Data Extraction and Critical Appraisal

We utilized guidance for qualitative synthesis [49] (p. 169) and [51] (Appendix A) to extract and appraise the articles. A study characteristics table (See Table 1) was devised to identify information for each of the articles systematically. In line with CIS methodology, articles were not excluded based on quality, but the appraisal process aimed to support our understanding of the studies. The first author completed all data extraction and appraisal. These processes were also conducted and used for comparison by the last (*n* = 13), second (*n* = 3) and third (*n* = 3) authors. The research team discussed and reflected on the appraisal and content of the articles, and this information informed aspects of the interpretation within the synthesis [45].

### 2.4. Analytical Procedure

The CIS analytical procedure was informed by that undertaken in primary qualitative research (informed by grounded theory [52]). The analytical process (Appendix A) included a detailed inspection of the articles and subsequently extracting the articles findings (primary data) and discussion sections (secondary data) as the evidence for reanalysis. New, inductive coding of evidence commenced (which included open, axial, and then selective coding), followed by data categorization and seeking to identify reoccurring themes, relationships within, and critical interpretation of meaning, consequently developing synthetic constructs and the development of a new theoretical framework. We employed a dual saturation process [53] to advance this qualitative analysis (See Figure 1). To ensure a contemporary and current analysis of the topic, the initial search was repeated to seek new, more recent publications. Subsequent analysis occurred on these articles to explore the new data for additional inductive coding and themes (seeking data saturation). In addition, to ensure a higher level of theoretical generality, theoretical saturation (i.e., theoretical categories were adequately developed presenting ‘theoretical completeness’ [54]), we employed a deductive analysis seeking to confirm and refine the existing theoretical model developed from any new articles [55].

Reliability and validity were ensured throughout analysis via constant comparative methods to aid theory generation. Investigator triangulation was adopted throughout, as the research team regularly discussed and reflected on the review process and analysis. The first author developed reflection notes while screening, extracting data, developing initial analytical constructs, and discussing them with the last author. Mitigation of qualitative researcher bias occurred throughout the analytical process. For example, analytical interpretations of data were challenged through discussions across the multi-ethnic research team. The research team comprised: a SA female (first author), who had a research interest in SA health and T2D (PhD), and members of her family who lived with T2D; a male Black senior lecturer (PhD) with expertise in diabetes research (2nd author); a white female Reader in Applied Psychology (PhD) with expertise in qualitative research methods (3rd author); a male SA researcher with interest in SA health and T2D research and community practice (PhD) (4th author); and a White female Health Psychologist, Reader in Applied Health Psychology (D.Health Psyc.) and Project Lead with expertise in clinical diabetes and qualitative research methodology (final author).

## 3. Results

### 3.1. Search Results and Article Appraisal

See Figure 1 for a PRISMA diagram of the search process. The initial search yielded 44 articles. Subsequently, 23 articles were removed as duplicates. The remaining 21 articles were subject to systematic screening. At this stage, seven articles were removed based on the title alone (three recruited a non-UK sample, one included mixed ethnic group, one focused on heart disease, and two were abstracts). At the abstract stage, seven further articles were removed (three explored non-self-management practices, three recruited a non-UK sample, and one used a mixed sample, including patients with Type 1 or Type 2 diabetes). The initial database search yielded seven papers for retrieval. Following the full-text screening, two of these were excluded (one a review, one explored other comorbidities). At this stage, five articles were eligible for inclusion in the review [56,57,58,59,60]. A comprehensive berry-picking process [50] found an additional 14 articles for retrieval [61,62,63,64,65,66,67,68,69,70,71,72,73,74]. Thirteen of these articles were eligible for inclusion. Two of the articles [64,65] were published versions of studies included within another included article (a PhD thesis [60]), and therefore to avoid duplication, the peer-reviewed publications were included in the review [64,65], and the thesis was removed [60].

To ensure inclusion of contemporary literature and working towards theoretical saturation, the search process was repeated in May 2020. This second search process yielded two new articles [75,76]. In summary, the total number of articles included within this CIS was nineteen [56,57,58,59,61,63,64,65,66,67,68,69,70,71,72,73,74,75,76]

### 3.2. Description of Studies

Table 1 presents the study characteristics for the nineteen included articles. The nineteen studies involved 536 participants, with a representative age range from 18 to 84 years old. Of the data available, 43% of the participants were female (three articles did not report gender split [59,64,68], and one article reported a total sample of n6 as mixed gender, we assumed this sample was 50% female [70]). Overall, the sample represented a diverse heterogeneous SA population in the UK. Of the total sample, participants were identified as 25.93% Bangladeshi, 25.75% Pakistani, 18.66% Indian, or 27.24% representing the aforementioned groups as one, 2.43% were identified from another subgroup (such as Sri-Lankan, Nepalese) or listed as South Asian only. The studies were mostly conducted across England [59,63,64,65,66,67,68,69,71,72,73,74,75,76]. However, a small sample was conducted in Scotland [56,57,58,70], and in one study, some participants were recruited from England and Wales [61], although no studies were identified from Northern Ireland. The articles employed the interview or focus group methodology for data collection, and the analyses were mostly articulated as grounded theory or thematic analysis, although some methodologies were not transparent. Five of the articles (representing 25% of the participant sample) were published after 2014 and have not been evaluated in previous reviews [39,40].

The articles’ quality (Appendix A) highlighted pertinent barriers and challenges in diabetes self-management in the UK South Asian population. Overall, the articles varied considerably in their description of the process of data analysis. The earliest studies [66,71,72] lacked clarity in reporting analytical procedure, whereas more recent studies reported more systematic application and a sophisticated description of methodological and analytical-qualitative processes. We considered article quality as substantial in nine studies [56,58,59,64,65,70,73,75,76], moderate in five [57,61,63,67,68] and limited in five [66,69,71,72,74].

## 4. Analysis

### 4.1. Critical Interpretative Synthesis

In keeping with CIS principles [45] (and our application of grounded theory), we developed a new higher-level analysis, which goes beyond the findings within the original studies, and is explicitly designed towards new theory generation.

The analysis presents a new overarching synthesising argument (cultural conflict) managed by two sub-arguments: (1) decisional conflict for self-management behaviours and (2) management strategies and factors influencing conflict. These arguments are developed from seven synthetic constructs (themes developed from the analysis of articles). Table 2 indicates which studies offer evidence to support each synthetic construct through the participant raw data (quotes, primary data) or the article authors’ interpretations (secondary data). Please see appendices for Appendix A, which presents selected examples of extracts from the articles as (raw data) evidence to support each construct.

The following presents an analytic commentary of the synthetic constructs devised from the original articles. Subsequently, we present a description and illustration (Figure 2) of the overall synthesising argument and conceptual model developed to organise and relate the analysis and constructs to each other.

### 4.2. Synthesising Argument 1: Decisional Conflict for Self-Management Behaviours

Self-management behaviours included those related to dietary, physical activity and medication management. Patients reported understanding health care advice delivered from their diabetes team. However, there were some contradictions in T2D knowledge and application to self-management (as evident in patients’ descriptions of procedural knowledge regarding medication management or adapting dietary behaviour, in line with medical advice). Patients considered information regarding self-management behaviours as rudimentary and generic. Patients did not explicitly recognise culture-specific adaptations in their care, and as such, SA patients felt disconnected from healthcare advice. Of importance was the overwhelming influence of a patient’s social-cultural influence on their self-management behaviours.

#### 4.2.1. Diet

Typically, patients lacked awareness of making a connection between T2D management and specific dietary behaviours [56,58,63,69]. Patients considered diet in relatively simple terms, such as reducing consumption of traditional sweets and foods, such as ‘Roti’ [58,61,65]. Patients demonstrated misconceptions of dietary and medication recommendations. For example, they did not comply with national dietary guidelines [77] for fruit and vegetable consumption. They believed intake of fruit and vegetables would significantly increase glucose levels, and thus consuming these foods would have a detrimental impact on their T2D [69]. Patients also acknowledged high blood glucose levels as problematic. Hence, they made behavioural decisions such as not eating but only drinking to reduce glucose levels or to take less medication [56,58]. However, most often, patients made these decisions independently and did not discuss their behavioural decisions with their health professional. Cultural identity influenced patients dietary habits, which negatively affected their T2D self-management. For example, the SA traditional breakfasts comprised a strong thick cup of tea with milk and sugar with biscuits [63]. Typical lunch and dinner meals were described as chapatti, curried vegetables, dahl and rice [69,72]; such foods were considered staples in the patient’s everyday diets. SA diets were compared to the traditional UK diet [63,72]. There was some awareness of ‘bad eating habits’ and their effects on T2D outcomes.

However, certain foods were considered part of SA culture and helped to define SA identity; thus, they were challenging to let go of or adjust [58,71]. Patients made comparisons; for example, SA patients categorised White British food as bland and tasteless compared to SA curries, rice, and chapattis. Not eating SA cultural food was suggested to be detaching themselves from their SA culture [58,72]. Specific tastes and flavours were essential to the SA population, and the use of oil and ghee were considered essential ingredients used in cooking methods [58,63,67,69,74].

Moreover, certain foods were perceived to be better for the body if cooked in a culturally adapted way [63]. Although alternate cooking methods, e.g., boiled, baked, and grilled, were acknowledged, there was hesitation to change cooking methods [58,63,69]. Patients reported being disconnected from health professionals’ dietary advice if the importance and relevance of SA cooking methods and tasting preferences were ignored [58,61,63,69,71,72,75,76]. It is possible to resolve conflict, concerning cultural identity and diet, by health professionals assessing a SA patient’s cultural needs and attitudes of importance towards diet and specific foods [58,63,72]. For example, it could be mitigated by explicitly educating SA patients on culturally appropriate healthy eating practices [58,66,72,75,76], although this was not typical in the reporting of advice.

#### 4.2.2. Physical Activity (PA)

SA patients did not consider exercise as part of daily lifestyle behaviour as an acceptable social or cultural norm [57,58,59,75,76]. SA women reported staying at home and feared going out to exercise due to ‘bad experiences’ related to their T2D (e.g., fear of passing out in public and being negatively judged by others) [57,58]. Women were extraordinarily unlikely to take part in PA [57,59,73]. Explanations for not engaging in PA referred to practical issues, such as a lack of time, working anti-social hours, conducting household responsibilities, or not being able to find single-sex exercise classes [57,66,69,73,74]. Taking ‘time-out’ to engage in PA was considered ‘selfish’ [57]. To be given ‘permission’ to engage in PA, patients sought endorsement or encouragement from significant others. Of particular importance was the endorsement or promotion of specific PA from religious centres, considered both conventional and appropriate [70,74]. Health professionals should be mindful of religious nuances when recommending PA; for example, participants from different religious backgrounds reported varying PA beliefs. Bollywood and Bhangra dancing was highlighted as explicitly linked to the Sikh religion [78], and it is noteworthy that there are differing and controversial views regarding listening or dancing to music for Muslims (which may be considered al-haram, not permitted).

#### 4.2.3. Medication

For many patients, T2D was managed (initially) via lifestyle only. SA patients reported apprehension about taking medication. Prescription medication (e.g., metformin) was thought as unnecessary or could aggravate their health further or may confirm their role as a ‘sick person’ [56]. These perceptions have been endorsed by SA countries, where taking medication is perceived to have long-term damaging effects if taken over a long time [56,63,64,67]. Other beliefs surrounding T2D medication were evident across the analysis: such as perceiving it to be detrimental to their health, only taking prescriptions for instant relief (such as when recognising their blood glucose readings were ‘high’), forgetting or not prioritising medication, misunderstanding side effects or symptoms (e.g., taking all medications if ‘blood glucose reading was high’) and perceiving diabetes as an onset for other conditions [56,73]. Patients did not report opportunities for raising concerns regarding medication beliefs or opportunities to explore culturally relevant behaviours related to prescription drugs.

SA patients who were prescribed medications reported self-adjusting dosage without seeking medical advice [56,58,63,64,75,76]. Often, patients did not seek medical advice, and such behaviour could occur over an extended period and never be raised with their health professional [56]. Fasting behaviour can significantly impact T2D management, and patients may not have sought medical advice or indeed not informed their health professional of their intention to take part in prolonged fasting behaviour.

The decision to use ‘alternative medicines’ was common for SA patients [56,61,68,72,73,74]. Acceptance of alternative medicines has been influenced by cultural practices within countries of descent [68]. Many perceived that the alternative medicines were more effective than (Western) prescribed medicine [56,61,68,72,73], and this was not something they discussed during health care consultations. For example, food items such as Karela (the most popular), Guar, Tindora and Methi leaves were considered to aid glucose control if eaten regularly. It was reported that SA patients used these foods as therapeutic agents [61,72,73,74].

Furthermore, it was disclosed that family and friends (social pressure) encouraged patients to take these alternative medicines as they believed them to be more effective than traditional prescription medicines [61,73]. In the UK, alternative and herbal medicines are not regulated and are not prescribed or recommended [79,80]. The lack of acknowledgement of alternative medicines highlights a direct conflict between healthcare practice and cultural beliefs. Most patients would not discuss the use of herbal medicines with their health professional, potentially impacting their T2D [56,68,73].

Despite the broad acceptance of alternative medicines across patients, some believed that health professionals’ advice was sufficient, and there was no need to take herbal medicines. Some acknowledged that UK health professionals were trustworthy prescribers (they had no financial gain compared to health professionals in their ‘home countries’ where they had to pay for care) [56,68,73]. Nevertheless, this reanalysis highlights healthcare professionals’ need to explore possible beliefs in alternative medicines and patients’ potential plans for use.

### 4.3. Synthesising Argument 2: T2D Management Strategies and Factors Influencing Conflict

A diverse ethnic and religious background was reported within the articles; most participants had originated from their (home) country of origin and held their cultural identity as very important to them. Social and cultural roles played a significant part in an SA’s lifestyles [57,58,64,65,66,67,68,72,73,75,76]. Frequent attendance at social gatherings, events and partaking in several religious festivals (such as Eid for Muslims, Diwali for Hindus and Sikhs) was deemed necessary [57,58,64,65,66,67,72,73,75]. At these events, meals would be served, consisting of foods high in fat, sugar, salt, and low fibre [57,58,64,65,66,67,68,72,73,75]. Despite being aware of the effects of consuming such foods, individuals felt obliged to abide by cultural expectations (social pressure). This cultural-social activity would be prioritised over and above their needs to consider their T2D self-management behaviours, reinforcing social norms [57,58,64,65,66,67,68,72,73,75]. Refusal to eat such foods or consume their own foods at social and special events would be deemed disrespectful and offensive to others [57,58,63,64,65,72,73].

#### 4.3.1. Social Conflict

Overall, women were found to lack family support, and their responsibilities often hindered their T2D management [57,58,59,64,68,69,73]. The ‘women’s role’ was typically to stay at home, cook, clean, and take responsibility for the children [57,59,64,67,68,69,73]. Although social events with families were frequent, taking part in family-based PA was not practical [69]. Dietary behaviour was particularly problematic for women. SA women were considered responsible for family cooking [57,59,64,67,68,69,73]. The female participants reported attempting to integrate healthy options into their family meals (e.g., separating individual portions before adding sugar) [58]. Family preferences and adherence to traditional cooking meals and cultural foods took priority before implementing specific dietary advice to support diabetes management [59,64,73]. Males rarely acknowledged input in family meal preparations [58,59,67]; thus, their food consumption was often determined by another (wife/mother/daughter).

Motivation via family support was fundamental to promote effective self-management behaviours (especially those relating to dietary behaviour), which helped implement T2D management [65,73,74,75,76]. Studies suggested various contributing factors, which facilitated conflict with a patient’s culture, such as cutting down on staple dietary foods, including chapatti and rice. These foods held significance to SA culture and suggesting a change in food consumption created (internal/external) conflict when making choices to their T2D self-management [57,61,64,65,69,72]. When this conflict was resolved, and the patients could accept their T2D as a priority behaviour, they made adaptable changes to their lifestyle, in-line with diabetes advice, and improved their self-management [57,58,64,65,70,72,73].

#### 4.3.2. Religious Obligations

Religious beliefs were found to influence self-management behaviours, such as ‘informing food choices’. Religious beliefs were explicitly pertinent to the patients aligned with Muslim religion [63], who adhered to religious rules relevant to food intake, e.g., consuming only halal (blessed food) and not consuming haram (forbidden) foods [81,82]. However, these rules’ relevance and practicalities were typically not acknowledged by health professionals when discussing dietary recommendations [63].

Religious beliefs influenced self-management through ‘religious obligations’. For example, Sikh religion patients explained that the Sikh temple held weekly meals serving traditional foods, such as chapatti and curries. Patients reported awareness of the ingredients and cooking methods used in these meals. They acknowledged the possible subsequent adverse effects of such foods on blood glucose levels (affecting T2D management). However, the food was deemed ‘essential’ to consume, as it would be disrespectful not to participate [58]. Hence, social pressure to engage and consume foods in these instances was reported.

Religious obligations, such as fasting, appeared to supersede T2D responsibilities. For example, Hindus reported ‘fasting for God’ as a priority rather than dietary influences on their diabetes [66]. For Muslims, fasting periods were compulsory elements to demonstrate adherence to religion—for example, participation in Ramadan (a 30-day fasting period). However, if a person were, for example, pregnant or had an illness, they may be legitimately exempt from fasting behaviour [58,63,64]. However, this practice was controversial, and for some with diabetes, they did not consider T2D as an illness to exempt them from fasting behaviour.

Religious beliefs overpowered diabetes management as patients reported that having diabetes was God’s will, so they had no control of their condition [57,64]. They were accepting diabetes as God’s will, promoting a sense of helplessness in their long-term T2D outcomes. Thus, any changes made in their behaviour towards self-management would ultimately have no impact, so the participants used this to explain not engaging in the recommended self-management behaviours. It is noteworthy that current NHS provision, promoting T2D education, may not address SA patient beliefs (such as T2D is caused and influenced by the will of God and sense of helplessness). The synthesis suggests that where current healthcare services fail to explore a personalised approach (which allows individuals to fully express and explore their lifestyle, cultural and religious beliefs), this appears to prevent the patient from successfully managing their T2D behaviours.

#### 4.3.3. Healthcare Delivery

Health professionals were acknowledged as vital influencers in patient’s diabetes care [56,61,65]. Health professional’s advice and prescriptive recommendations were perceived to be outside of a patient’s control. Hence, at times, patients reported helplessness [65]. Across studies, various aspects of misunderstanding health professional advice were apparent [58,59,67,71,75] but often not acknowledged or challenged.

Patient’s relationships with the health professional were respectful, and patients considered the health professional as the expert in their job role. Hence, they would not feel comfortable challenging recommended care. Older SA patients reported preconceived ideas of how health care systems should work due to the experiences they had in their ‘home’ countries (e.g., paying for treatment, they expected the best care, in contrast to the UK NHS as a free service). As such, SA patients often considered the NHS as a bit of advice and interpreting service. Although there was recognition of NHS health professionals as ‘trustworthy prescribers’ (given they had no financial gain compared to health professionals in ‘home’ countries), these historical viewpoints and perceptions influenced engagement with their care, often leading to non-compliance with recommended guidance [56,59,61,67,71].

Patients described a conflict between managing their lived experiences of T2D against health professionals’ recommended advice. Patients considered that the health professionals gave little acknowledgement to their cultural relevance and specific social-environmental needs. For example, patients would be advised not to eat specific foods at social gathering or events. However, the patients interpreted this as a lack of cultural understanding, given it would be viewed as socially offensive not to participate in such a social gathering fully. The patients reported a sense of being judged negatively within their community [64]. Equally, social experiences would not be shared directly with the health professionals, through the fear that aspects of their ethnicity and SA community may also be evaluated negatively. In this essence, the patients are stuck between a decision of causing ‘social offence’ (which may have lasting social implications for themselves/family) or a lack of engagement with healthcare advice (which has potential but unknown implications for T2D outcomes).

#### 4.3.4. Health Beliefs, Language, and Literacy

Overall patients had a lack of understanding towards T2D [57,61,63,64,67,69,74], and were most likely to reject medical explanations for the onset of diabetes [61,63,67,69]. Some believed that T2D was caused to factors outside of their control, such as stressful events and worry [64,67,69,72,74] or the will of God/Allah) [57,64]. Believing that the causation of T2D was external and of limited relevance to their behaviours subsequently harmed their understanding and application of self-management strategies [58,61,65]. These beliefs were often reinforced by patients’ evaluations of information received (or not) from health professionals [58,61].

Moreover, patient’s perceptions about their future health influenced the acceptance of T2D as a severe condition. For example, the acceptance that ageing made illness unavoidable (inevitable) and uncontrollable [57,64,67] reinforced patient beliefs that self-management behaviours have limited effect. These beliefs were made by patients comparing themselves to older members of their SA communities, who may have lived with T2D for many years. Patients considered T2D as a social norm within their family and SA community. As a social norm, this acceptance reduced their perception of T2D as a health risk and influenced their decision to (not) implement healthcare advice.

However, whilst T2D was a social norm, cultural perceptions also influenced the participant’s health beliefs. To acknowledge a health condition such as a diagnosis of T2D was stigmatised and negative. In this regard, the concept of T2D was a social norm but not an acceptable status. Engaging in observable physical behaviour, such as injecting insulin, would negatively influence an individual’s evaluation of themselves, their community, and social evaluations [57,64].

Moreover, a misperception that is going home to ‘India’ would ‘cure’ their diabetes as T2D was attributed to the patient’s external locus of control (environmental factors such as living in London). Rather than accepting personal responsibility for their diabetes and engaging with their internal locus of control (e.g., the power to change dietary and exercise habits) [65,66,67].

Foreign language was an obstacle for patient interaction with health professionals [61,67,69,74,75]. Having language translators or an interpreter during consultations was suggested to eliminate communication difficulties. Although translators helped patients understand basic information, this approach limited health professional-patient interactions and subsequently reduced the opportunity for clarification and questions. Patients felt unable to comprehensively address their queries and uncertainty (perhaps due to time limitations as interpretation extended the consultation process or perhaps due to the patient receiving basic information from the interpreter but not having the confidence to ask questions themselves) [61,67]. It was reported that information was best understood if received from a doctor who conveyed messages in the SA mother tongue [61,68].

However, the synthesis suggests (foreign) language translation was just one aspect that reduced effective communication between the health professional and patient [75]. The analysis indicates additional difficulties with health literacy surrounding T2D care [75]. Health professionals’ communication, terminology, and explanations of T2D and healthcare decisions during consultations were challenging to understand, regardless of language spoken [58,61,72]. For example, terms such as ‘dieting’ and ‘diet-control’ have been poorly understood [58,61,69], diet control was assumed to focus on ‘sugar’ intake only [61].

## 5. Synthesising Argument—Cultural Conflict vs. T2D Management

This new analysis suggests patients reconstruct knowledge to manage their psychological (beliefs), behavioural (social influences), and cultural (including religious) conflicts, which impacts their decisional conflicts to engage with T2D self-management practice and health professional advice (un)consciously. SA patients face conflict aligning to their cultural identity versus engaging with medical advice to change their lifestyle behaviour. Our analysis suggests that participants renegotiate their cultural acceptance to resolve such conflict and engage in T2D self-management behaviours. Alternatively, they do not internalise the medical advice and thus allow their cultural identity to supersede T2D as a priority. Those who were unable to balance out this conflict were more likely to default towards cultural identity and continue to align to the cultural preferences rather than health professional guidance and thus not change lifestyle behaviours for self-management.

Figure 2 presents the conceptual model of the synthesising argument in which the spring represents the conflict that exists and that which the patient must learn to manage.

## 6. Discussion

### 6.1. Interpretation

A key feature of this interpretative synthesis is that the analysis was critical, whereby it questioned the published literature on the concept of SA patients’ beliefs which impacted on engagement with T2D healthcare and thus self-management behaviours. This review was not an integration of existing evidence but an interpretation that creates a new understanding and theoretical explanation.

The CIS framework suggests that SA participants prioritise their cultural, social, and religious identity, which inform and strongly influence their behaviours and daily lifestyle. The link between the concept of identity and self-management is that health professional advice appears to create internal (decisional) conflicts for these SA individuals. These conflicts, if not managed well, interfere (often negatively) with their T2D self-management.

Conflict Theory [83] explains how people cope with the dilemmas of decision-making. Decisional conflict refers to a person’s reaction when deciding whether to accept or reject an action. Symptoms include hesitation, emotional distress, and feelings of uncertainty. However, this reanalysis suggests that SA patients may not recognise such decisional conflict. Decisions may not be conscious, and the patients may make unconscious habitual, culturally enforced decisions rather than renegotiating behavioural choices. Decisions based on a sense of control and ability to respond, rationalise, and negotiate behavioural changes with their SA family, friends, social and religious networks may be difficult, and patients need support in doing so. However, to make informed decisions and balance out any conflict, patients must also have a sound understanding and knowledge base surrounding T2D. This reanalysis suggests that many SA patients hold skewed illness representations and lack knowledge and skills to understand T2D and behavioural management (such as dietary, PA changes, or taking prescribed medications).

There is a clear need for services to address a patient’s health beliefs regarding the cause and management of T2D, and such health beliefs may also exist in patients from other ethnicities. However, the concept of social norm is particularly relevant to the SA population, given this is a highly prevalent condition within the SA community. These social norms reinforce (negative) beliefs that, unless addressed and mitigated, will influence a SA patient’s acceptance or implementation of advice received from health professionals. Moreover, as a result of SA expectations of healthcare delivery, patients may not openly seek clarification or question advice from health professionals. Although low health literacy relating to T2D more generally was apparent, foreign language may add to communication difficulties. Assessing, re-assessing, and mitigating social-cultural influences of patients’ health beliefs should be an ongoing aspect of T2D healthcare delivery.

Previous research has considered practical recommendations for the delivery of T2D care and interventions. Indeed, a review [84] highlighted the role of considering individual differences in intervention delivery. However, that review did not consider ethnic disparity within healthcare, and indeed in doing so, how cultural and ethnicity influenced patients’ experiences of such recommended interventions. Previous reviews [39,40] have not synthesised findings to consider SA patients’ underlying psychological processes; we suggest that this new synthesising argument has considered SA patients’ needs from a psychological rather than a practical (nursing) based response. This review highlights how the complex psychological processes and interpretations of SA patients’ information may not be evident to health professionals around them. The conflict, maybe, hidden from the health professionals or, indeed, unknown directly to the patients themselves. These CIS findings suggest that participants prioritised cultural identity above individual health needs; patients managed the conflict by comparing their social norm (other SA behaviour) and prioritised lifestyle that superseded health (T2D) behaviours.

### 6.2. Relevance to Clinical Practice

Our synthesising argument and supporting model (Figure 2) promote novel ideas for delivering T2D care for SA patients. Patient education and ongoing care need to explore patient’s conflict-related to influencing factors such as cultural and religious expectations, social norms, and social comparisons. Health professionals need to consider patients health beliefs as influenced by cultural and religious knowledge, such as investigating attitudes towards prescription medications and alternative medicines. Health professionals must go beyond communication and (foreign) language barriers and further consider health literacy and understanding of T2D management. Cultural expectations affect people’s help-seeking. For example, the impact of T2D being a social norm within the SA community, thus normalising symptoms, and engagement (or not) with health professionals or implementation of self-care behaviours and the social expectation to continue with cultural and religious obligations and behaviours. People need to identify and evaluate their symptoms, negotiate relationships with health professionals and consider how they may need to negotiate support with their family and social network. These tasks may require resources from both the patient and the health professional unless these psychological needs are explored and entwined into T2D ongoing care. SA patients will likely continue to experience conflict, observed via difficulties with health service engagement, and implementing the recommended self-management behaviours.

It is noteworthy that we are not advocating that services should differ significantly for SA patients than other patients with T2D. This review highlights the complexities for SA patients when balancing the conflict between their cultural identity and T2D. However, health professionals must be trained, supported, and set up to manage SA patients (and others) expectations of health care. Health Professionals should explore and mitigate negative health beliefs and enable patients to manage social-cultural influences that impact their self-management behaviours. We do not wish to imply that hidden or unknown health beliefs do not exist for other patients with T2D; we highlight that the cultural-ethnic belief systems that inform SA patients’ cultural identity are the most prominent beliefs that impact their engagement and implementation of healthcare advice.

### 6.3. Ethnic Minorities, COVID-19, and T2D

It is essential to acknowledge the implications of these findings relevant to the new 2020–2021 COVID-19 pandemic [85]. The current data report (Jan 2021) that COVID-19 has been responsible for 2.24 million deaths worldwide [86] and 112,660 deaths (3 February 2021) in the UK alone [87]. It is particularly noteworthy that within the current climate of the COVID-19 pandemic, evidence suggests that people with T2D have been disproportionately affected by COVID-19, with growing evidence of higher mortality and morbidity [88]. Moreover, people who are categorised as an ethnic minority are at an additional risk of COVID-19 complications, and this risk also increases significantly for those with a diagnosis of T2D [89,90,91]. The emergency response to COVID-19 worldwide has led to numerous public health controls. These have included the wearing of facemasks, social distancing, quarantine, and lockdown. These measures have affected daily living, social interactions, the workplace, and healthcare. Healthcare services have adapted (and continue to do adapt) their delivery style and refocused their resources based on emergency need. Specifically, a diabetes service in the UK [92] reported having to suspend non-emergency care and cancelling all routine diabetes appointments, including those in community services. The services are operating under new care pathways and clinical guidelines [93,94]. Patients categorised as urgent received a telephone/video consultation, and a telephone consultation line was implemented for general support. In Taher and colleague’s clinic, they acknowledged their Muslim patients’ needs and followed the newly established guidelines [95] for supporting Ramadan during this period.

However, we report that (pre-COVID-19) SA patients seek to manage multiple conflicts between their cultural, social, and religious identities against their healthcare advice and T2D self-management behaviour. Considering the changes associated with COVID-19 and T2D care, the relationships with healthcare providers, access to information and support may be further negatively influenced by the COVID-19 pandemic. Moreover, evidence suggests that people from black and ethnic minority groups are hesitant about healthcare advice, showing resistance to accepting the COVID-19 vaccine [96]. This presents a new emerging picture of COVID-19 influencing the cultural-social conflict with healthcare and emphasises further inequalities in healthcare for patients with T2D, especially from an ethnic minority background, such as SA. While the evidence synthesised within this review does not include studies conducted during the COVID-19 period, we would suggest that the conflict between SA T2D patients will be exacerbated during this time. The patient’s ability to manage and resolve these conflicts will be more challenging, and as such, our model would predict that SA patients will revert to their social norm and culturally accepted and promoted approach. Public Health England published a report which acknowledged recommendations from stakeholders who called for the need to “accelerate efforts to target culturally competent health promotion and disease prevention programmes for non-communicable diseases…including diabetes” [97]. However, this report prints a clause that “the requests for action from stakeholders and do not represent the views of PHE” (p. 48), and the report lacks guidance on how to tackle or implement the recommendations. We propose that a starting point would be to consider the new conflicts that COVID-19 has brought forward into SA patients’ T2D care and how these further interfere with self-management behaviours. Healthcare providers will need to start adapting service further to refocus and account for cultural identity and health inequality, supporting patients in managing social, cultural, and religious conflicts, and they will need support, time, and resources to do so.

### 6.4. Future Research

Findings from this CIS are significant regarding future research. This synthesis highlighted that previous research has focused on barriers and enablers to T2D self-management. Future research should explore the practical application for proactive ways to facilitate the factors that could help the SA population better self-manage. The synthesized findings and model could be utilized for patient and public involvement with SA patients to discuss specific service improvements and new interventions. Moreover, the findings could aid care quality discussions with key stakeholders, such as healthcare providers working in diabetes care.

It is important to highlight that there was limited evidence reviewing the advice and delivery of care regarding increasing PA; hence, it would be beneficial for future research to explore SA patients’ lived experiences receiving advice and engaging in PA. A worldwide systematic review on the effectiveness of diabetes self-management programmes targeted towards ethnic minority groups reported that only 65% of programmes included physical activity education elements. The review reported that interventions were more successful in improving dietary behaviours than promoting PA or medication adherence [98]. Another review [99] evaluated a diverse pool of interventions that targeted SA patients with T2D. It highlighted that interventions that were more culturally adapted were more successful overall. Specifically, regarding PA, of those studies conducted in Europe, PA was included as an educational (knowledge only) component of the interventions (rather than the practical and supportive approach as per the studies conducted outside of Europe, specifically in India). It was suggested that PA might not be a prioritised component of T2D care delivery, which could reinforce SA patients’ perceptions of the (none) importance of daily PA or (limited) relevance to their T2D self-management. However, whilst both quantitative reviews [98,99] included randomised controlled trials, the evidence focusing on PA is limited and much is of low quality. Moreover, the studies included in these reviews do not indicate how or if the new interventions were implemented in clinical practice. Nevertheless, the findings presented would be supported by the synthesising argument presented within this review, although this synthesis explores the reality of T2D self-management for SA patients and highlights the need to improve the support and advice available to increase their PA.

### 6.5. Strengths and Limitations of the Review

This review allowed for articles from different qualitative methodologies and epistemology to be analysed together, resulting in a coherent, theoretically informed model. Specifically, five articles [66,69,71,72,74] had reduced quality appraisal for their lack of analytical detail, and the analytical process applied within this review has helped to mitigate and validate these data. CIS supports data extraction relevant to the research question to ensure a focused and critical approach to a new interpretation of findings [46]. All synthetic constructs were supported by the evidence presented from at least five of the included articles (see Table 2); none of the constructs here were dependent solely on low-quality articles. This review was supported by a rigorous quality process, ensuring consistency across the research team in the agreement of article selection and analytical processes (in line with GT methodology).

The studies included within this CIS referred to patients throughout as SA. However, the heterogeneous nature of the SA population was overlooked or simplified in many. Some studies [54,55,56,57,67] acknowledged limitations in the SA sample recruited; such limitations mainly occurred due to logistical research issues such as including participants based on speaking English (or other restricted SA languages). None of the studies acknowledged the complete heterogeneous nature of the SA population in the UK. However, this CIS has brought together participants across a range of studies, and thus the reanalysis and reconstruction were able to develop a more realistic and heterogeneous consideration of the SA population experiences of T2D self-management behaviours.

We acknowledge that methodological issues may have contributed to the limitations of this synthesis. The literature searches utilised a range of databases, and additional berrypicking [50] approaches were applied. However, the search may not have been fully exhaustive. The proposed synthesising argument and model are developed from post hoc retrospective accounts of participant’s evidence. Therefore, future research to evaluate real-time cultural conflict changes and their application to healthcare advice and subsequent T2D self-management behaviours is warranted. The evidence available within the articles focused mainly on diet and medication as the focus of self-management behaviours, less included commentary of physical activity in SA patients. It was not possible to differentiate the evidence from a gender perspective or separate the constructs into evidence for SA patients’ full-heterogeneous range. Finally, it could be considered a limitation that only English language papers were included or that the review focused on UK based studies only. However, given the heterogeneous nature of the UK SA population [4], and the unique nature of the UK National Health Service, combined with the social-cultural population mix within the UK [100], it was deemed plausible to explore articles from this perspective, although we recognise that the results of this review may not be transferrable to other countries. It is noteworthy that T2D prevalence has increased across the world; for example, in America and Canada, the SA population are at higher risk of developing T2D [101,102], and the prevalence and needs of SA patients in westernised populations have high rates of health inequalities [103]. Therefore, the findings of this article may be useful for consideration across the world to help explain cultural–health care conflicts and how to improve care delivery.

## 7. Conclusions

Our review was both critical and interpretative. Whilst being systematic in methods applied, it offers a new and reflexive approach to the previous evidence. The new analysis has been created directly from the reports of SA patients who have contributed to a range of qualitative articles. A new conceptual framework is offered to help understand the SA patient experiences and highlight service improvement and health professional practice considerations.

## Figures and Tables

**Figure 1 ijerph-18-02641-f001:**
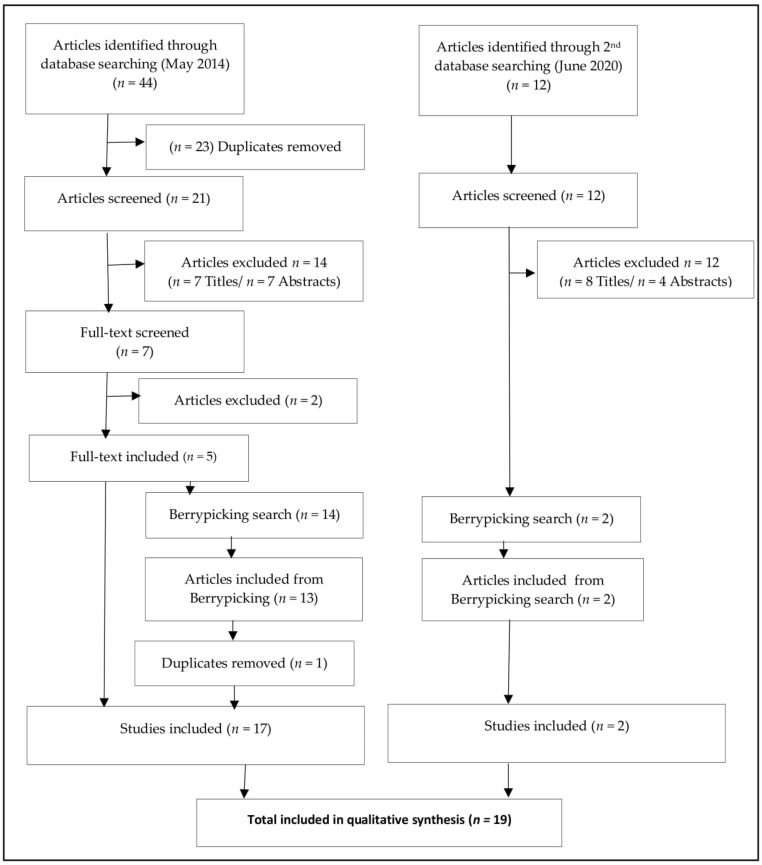
PRISMA flow-diagram of the search and screen process.

**Figure 2 ijerph-18-02641-f002:**
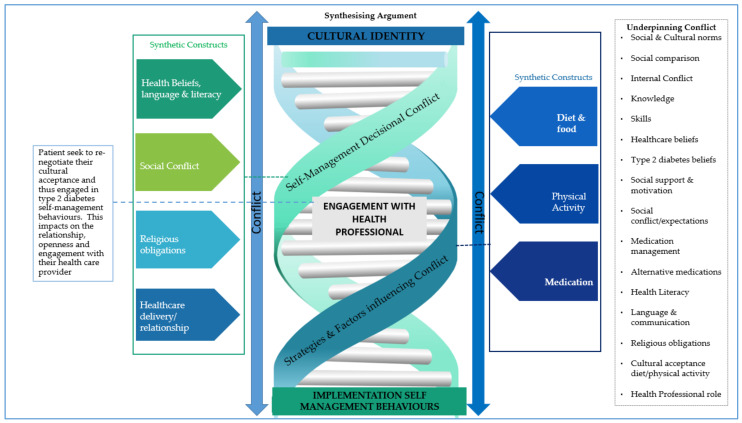
Model of synthesizing argument.

**Table 1 ijerph-18-02641-t001:** Study characteristics.

Reference, Author, Year of Publication	Purpose/Objectives	Sample Size	Location	Method of Data Collection	Data Analysis	Main Results/Key Findings
Khajuria and Thomas (1992) [72]	Explore the extent to which traditional beliefs about diet, health and diabetes, as described in the classics of Ayurvedic medicine, are held by Indian (Gujarati) diabetics in Britain.	28 Gujarati, Hindu, vegetarian, patients with diabetes	Diabetic clinics at two London Hospitals. (England)	Interviews	No clear indication as to which analysis applied, however, results are shown in themes	- Belief in traditional remedies and the role of food in the management of diabetes. - Traditional meal patterns, Patients eating habits. - Dietary advice received from the hospital
Kelleher and Islam (1994) [71]	To understand and describe how Bangladeshi people with diabetes attempt to integrate a traditional and religious rule-governed system of eating with modern medicine.	40 Bangladeshi, 25 males and 15 females, varying ages. 30 family members	One health centre in the Tower Hamlets district. (England)	Interviews and Observations	Preliminary analysis on a selected sample of *n* = 20, 12 men and 8 females. Unknown how the sample was selected for analysis.	- The development of self-help groups for Bangladeshi people will encourage them to talk about their experiences of diabetes and the treatment regimens in the contexts of their everyday Muslim lives.
Chowdhury et al. (2000) [63]	Report on the food beliefs and classification system of British Bangladeshis.	40 informants with diabetes, all immigrants from the Sylhet region of Bangladesh	Inner-city areas of Tower Hamlets, Newham or Islington in London. (England)	Audiotaped narrative, Semi-structured interviews,Focus groups,Construction of genogram, Pile sorting exercises,Structured vignette method,Feedback of preliminary constructs to focus groups,Study of patients’ general practice case notes	Anthropological analysis.No quotes from participants as evidence. Interpretative commentary with reference to the previous evidence base.	- Primacy of religious prohibitions. - Ethnic dietary customs imported from Bangladesh. - Modification of diet on immigration to the UK: impact of availability and affordability. - Classification of food: strength and digestibility dimensions. - Digestibility dimension. - Strong-weak dimension. - Role of cooking in modifying edibility. - Food, health and illness.
Bissell et al. (2004) [59]	Explore the relevance of a re-framed consultation with a small group of English speaking patients of Pakistani origin with a diagnosis of T2D	21 Pakistani origin participants	Two primary practices and one secondary care diabetes centre located in the northwest of England.	Semi-structured interviews	Grounded theory	- Themes identified: - Integrating the diabetic regimen, - Respondents’ experiences of healthcare interactions
Lawton et al. (2005) [56]	To explore British Pakistani and British Indian patients’ perceptions and experiences of taking OHAs.	32 patients of Pakistani and Indian origin with T2D.	Primary care and community sources in Edinburgh, Scotland.	An observational cross-sectional study using in-depth interviews in English or Punjabi.	Grounded theory	- Initial reactions to taking OHAs, - Perceptions of OHAs, Self-regulation of OHAs - Self-regulation strategies
Macden and Clarke (2006) [64]	Developed knowledge of the experiences of South Asian people with diabetes in the UK in relation to socio-cultural and dietary practices, religion and ageing influences on the perception and understanding of risks.	Ethnic health development workers, health professionals and 20 SA men and women with T2D.	North East England.	Focus group interviews and one-to-one interviews	Grounded theory	- Factors influencing risk perception - Weighing up risks
Lawton et al. (2006) [57]	Explore patient perceptions and experiences of undertaking PA as part of their diabetes care.	32 participants (Indian, *n* = 9; Pakistani, *n*= 23).	Five general practises in Edinburgh. (Scotland)	Semi-structured interviews	Grounded theory	- Lack of time: obligations to others - Fear and shame - Lack of culturally sensitive facilities - Climatic conditions - Comorbidities - Accounts of causation: perception of future health - Diabetes triggers an irreversible decline - PA can engender anxiety - Short term goals - I do enough already
Lawton et al. (2008) [58]	To look at food and eating practices from Pakistanis and Indians’ perspectives with T2D, their perceptions of the barriers and facilitators to dietary change, and the social and cultural factors informing their accounts.	23 Pakistanis (22 Muslims, one Christian) and nine Indians (four Hindus, five Sikhs)	Five general practices in Edinburgh	Semi-structured interviews	Grounded theory	- Information from healthcare professionals--- - Perceptions of SA foods: bad for health; good for self - Settlement, sharing and commensality - Strategies for passing: cutting out or cutting
Choudhury et al. (2009) [61]	Examine the understanding beliefs of people with T2D from the Bangladeshi community living in the UK.	14 Bangladeshis	Swansea or Birmingham (Wales/England)	Structured Interviews	Theme analysis	- Cause of diabetes - Preventing diabetes - Diabetes diagnosis - Management of diabetes - Information from health care professionals - PA - Information from family/friends and use of traditional medication - Diabetes education
Macaden and Clarke (2010) [65]	To analyse risk perception among older SA people with T2Din the UK.	Ten Health development workers, seven individual interviews with practitioners (three physicians, three nurse specialists and a dietitian); 20 interviews with UK-resident older SAs (nine men and eleven women) with T2D	North East of England.	Two focus group interviews with health development workers, seven individual interviews with practitioners	Grounded theory with its theoretical foundations drawn from Symbolic Interactionism.	- Perceptions of External Responsibility - Perceptions of internal responsibility - Control and influence
Jepson et al. (2012) [70]	Explore the barriers, motivators and facilitators to SA adults undertaking PA, with the broader aim of guiding the development of future interventions and services.	59 Bangladeshi, Indian and Pakistani participants 10 Key Informants	Urban areas of Scotland, Aberdeen, Glasgow and Edinburgh.	Focus group discussions with participants and semi-structured interviews with key informants,	Thematic analysis	- Types of PA people engaged in - Social interaction - Enjoyment of exercise - Mental and physical benefits - Leadership and role models
Gumbler (2014) [69]	Investigate whether there was a knowledge gap among SA women with T2D about diabetes development and management.	Six SA women	Warwickshire or in Birmingham	semi-structured interviews	No clear indication as to which analysis was used, however, results are shown in themes	- Diabetes knowledge - Diet - Physical inactivity and a healthy lifestyle - English fluency and language barriers - Potential improvements to the NHS - Other findings
Majeed-Ariss et al. (2015) [73]	To explore the effects of T2Don British-Pakistani Women’s identity and its relationship with self-management.	15 British-Pakistani women with T2D.	Teesside, England	Face-to-face semi-structured English and Urdu language interviews	Thematic analysis	- Perceived change in self-emphasised how British - Familiarity with ill health reflected women’s adjustment to their changed identity over time; - Diagnosis improves social support enabled women to accept changes within themselves - The over-arching theme Role re-alignment enables
Fleming, Carter, Pettigrew. (2008) [67]	To present the findings of a study which explored the influence of culture on (type 2) diabetes self-management in Gujarati Muslim men who reside in northwest England.	5 Gujarati Muslim men	Northwest England	A case-study approach	Topic and analytic coding	- The findings highlight that the complexity of life means that culture never exists in isolation but is one of the many factors that a man negotiates to inform his diabetes self-management.
Greenhalgh et al. (1998) [68]	To explore the experience of diabetes in British Bangladeshis, since successful management of diabetes requires attention not just to observable behaviour but to the underlying attitudes and belief systems which drive that behaviour.	40 British Bangladeshi patients	Three general practices in East London	Audiotaped narrative Semi-structured interview Focus group discussion Construction of genogram Pile sorting exercises Structured vignette method Feedback of preliminary constructs to focus groups Study of patients’ general practice case notes	No clear indication as to which analysis was used	- Body concept - Origin and nature of diabetes - Impact of diabetes - Diet and nutrition - Smoking - Concept of balance - Exercise - Professional roles - Diabetic monitoring
Duthie-Nurse (1998) [66]	The patients’ views of illness and how it was treated, with particular regard to diet	20 Hindu SA women	Diabetes Clinic, St Georges Hospital, South West London	Open and closed-ended interviews	No clear indication as to which analysis was used	- Consciousness of health and disease - Duties of sick women towards household and gods - Emotional stress, followed by ‘recharging’ through visits to family in India (and America) - Objective assessment of the interplay between environment and disease - Displacement and alienation as factors actively influencing the foregoing.
Patel and Iliffe (2016) [74]	To explore the influence of health beliefs and behaviours on diabetes management in British Indians, as successful management of diabetes is dependent on underlying cultural beliefs and behaviours.	10 British Indians	General Practice in North West London	Semi-structured interviews Pile sorting exercise	Thematic analysis	- Causal beliefs about diabetes - Adaptation of exercise - Use of alternative therapies - Modification of diet - Sources of information
Pardhan et al. (2018) [75]	To determine whether diabetes awareness and self-help barriers differ in South Asian participants of different demographic characteristics (age, gender, and literacy) with type 2 diabetes living in the United Kingdom.	35 participants, 26 were Pakistani, 5 Nepalese and 4 Indians	Five focus groups in community centres in Peterborough and one focus group in a research facility at Anglia Ruskin University, Cambridge campus.	Six Focus group discussions	Thematic analysis	- Knowledge /awareness about diabetes and its complications - Self-help and factors that influence self-help - Uptake of available healthcare services
Prinjha, Ricci-Cabello et al. (2020) [76]	Aimed to explore British South Asian patients’ perceptions and views with T2D on mobile health SMS text messaging to support medication adherence, aimed at the general UK population.	A diverse sample of 67 participants (Indian, Pakistani, Bangladeshi, and Sri Lankan)	Community centres in Leicester	Eight Focus group discussion	Thematic analysis	- Message content and design features - Language preferences - Family involvement - Different digital formats for different groups - Face-to-face groups for those who do not use digital devices

Abbreviations: T2D, Type 2 Diabetes; SA, South Asian; OHAs, Oral Hypoglycemic Agents.

**Table 2 ijerph-18-02641-t002:** Synthesising argument, synthetic constructs and summary of supporting evidence.

Synthesising Argument	Synthetic Construct (Themes Developed from the Analysis of Articles)	Supporting Evidence from Studies
Cultural Conflict vs. T2D Management		
Decisional Conflict for Self-management Behaviours	Diet and food (a) Understanding choices (b) Cultural identity (c) Cooking methods	(a) [56,58,61,63,65,69] (b) [58,63,66,67,69,71,72,74] (c) [58,59,61,63,69,71,72,75,76]
	Physical Activity (a) Social-cultural norm (b) Endorsement	(a) [57,58,59,66,69,74,75,76] (b) [57,70,74]
	Medication (a) Management (b) Alternative	(a) [56,58,63,64,67,73,75,76] (b) [56,61,68,72,73,74]
Management Strategies and Factors Influencing Conflict	Social Conflict (a) Family support and role (b) Motivational support	(a) [57,58,59,64,67,68,69,73] (b) [57,58,61,64,65,69,70,72,73,76]
	Religious obligations	[57,58,63,64,66]
	Healthcare delivery/relationship	[56,58,59,61,65,67,71,74,75]
	Health Beliefs, language and literacy (a) T2D knowledge (b) T2D timescale and social evaluation (c) Communication	(a) [57,58,61,63,64,65,67,69,72,74] (b) [57,64,65,66,67] (c) [61,67,68,69,72,74,75]

## Data Availability

All raw data extracted and synthesised in the present review are taken directly from the published articles (results and discussion sections) as listed in Table 1. Further details are available from the corresponding author upon request.

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
