# Peer review of "Cultural Identity Conflict Informs Engagement with Self-Management Behaviours for South Asian Patients Living with Type-2 Diabetes: A Critical Interpretative Synthesis of Qualitative Research Studies"

_ijerph, 2021, doi:10.3390/ijerph18052641_

Round 1
Reviewer 1 Report
The Systematic Review entitled "Cultural Identity Conflict Informs Engagement with Self-Man-2 agement Behaviors for South Asian Patients Living With 3 Type-2 Diabetes: A Critical Interpretative Synthesis of Qualita-4 tive Research Studies." has an interesting approach for publication in Int. J. Environ. Res. Public Health. Its describes a new conceptual that highlights the patient experiences and the needs for health professional practice in South Asian. But there are some questions of form that should be taken into account prior to consider this article for publication.
I attach the coments to author with the changes.
- The abstract should reflect a brief resume of the work, including the most significative results. Specifically, authors should note the results has been more relevant for the improvement of the patients and professional practice in South Asian.
- What is your objective to reflect that an update was made in the recruitment of information? Can it be unified?
- Flow-diagram should be simplified to improve its interpretation.
- The figure 2 is very extensive in some sections, I recommend the use of shorter items.
- Reconsider the format of table 1, it can be considered more appropriate for a meeting
- I recommend reviewing the format of the references, specifically those corresponding to book chapters.
Author Response
Reviewer 1
Response:
Thank you for the constructive commentary. We have addressed each of your comments as follows:
- The abstract should reflect a brief resume of the work, including the most significative results. Specifically, authors should note the results has been more relevant for the improvement of the patients and professional practice in South Asian.
We have improved the abstract to reflect this recommendation.
- What is your objective to reflect that an update was made in the recruitment of information? Can it be unified?
The review and synthesis was conducted inline with CIS methodology. We have added further clarification on the process of saturation within section 2.4. Analytical procedure
- Flow-diagram should be simplified to improve its interpretation.
Thank you for the comment, the prisma diagram is in line with CIS instructions to represent the repeated process however we have simplified the diagram.
- The figure 2 is very extensive in some sections, I recommend the use of shorter items.
We acknowledge the complexity of the figure, although believe that this does justice to the complex nature of the analysis.
- Reconsider the format of table 1, it can be considered more appropriate for a meeting
Initially we had considered placing this table in the appendices although on review of other manuscripts it seems these type of tables are standard in main text and so we have included the study characteristics as standard in this table. We will take a steer from the publishers if they would wish this to be moved into the appendices.
- I recommend reviewing the format of the references, specifically those corresponding to book chapters
The references were processed from endnote, and we have now reviewed the formatting thoroughly
Reviewer 2 Report
This is a useful review on a very important subject in the diabetes community. I have some comments to further improve the review:
Methods
- It would be useful to mention that grey literature would be excluded (if this was the case) and also if any type of qualitative design would be eligible.
- You should mention if quality appraisal was a criteria for inclusion - also line 242 introduces CASP and it might be better to mention this in the methods sections first
- Was the synthesis at the level of quotes (so more inductively) or was this at the level of the themes extracted by the original authors? (from line 177)
- Clarify the process described in line 185 to 187 regarding data saturation based on existing literature
Results
- It might be useful to divide the bulk of writing in the first results page by headings such as "Description of studies" and "Appraisal of studies" for easier reading
- Table 1 - this depends on the level at which you extracted data (quote level or pre-generated theme level), but I wondered if it would be useful to have a column for themes extracted, column for a supporting quote from that study, and a column for the collective interpretation by the author? Perhaps this might all go into a new table? So you would have a study characteristics table and a table of the extracted findings across each study? Then the final table would contain the synthesised findings. It would just tease out the analytic process a little more and help justify your synthesis.
Discussion
- Future work might also use the synthesized findings and model to engage individuals living with diabetes from a South Asian culture in patient and public involvement research and participatory action research to discuss future steps. Similarly discussions with other key stakeholders such as diabetes healthcare providers.
Author Response
Reviewer 2
Response:
This is a useful review on a very important subject in the diabetes community. I have some comments to further improve the review
Thank you for the constructive commentary. We have addressed each of your comments as follows
Methods
- It would be useful to mention that grey literature would be excluded (if this was the case) and also if any type of qualitative design would be eligible.
Thank you we have clarified these points (line 150/157)
- You should mention if quality appraisal was a criteria for inclusion - also line 242 introduces CASP and it might be better to mention this in the methods sections first
Thank you we have clarified these points, CASP was already referenced in the Data extraction and critical appraisal section as well.
- Was the synthesis at the level of quotes (so more inductively) or was this at the level of the themes extracted by the original authors? (from line 177)
Thank you we have clarified the inductive nature of the analysis
- Clarify the process described in line 185 to 187 regarding data saturation based on existing literature
Thank you further information has been added to aid understanding of data and theoretical saturation applied.
Results
- It might be useful to divide the bulk of writing in the first results page by headings such as "Description of studies" and "Appraisal of studies" for easier reading
Thank you, we have added the words – description of studies as a subheading, and Analysis- Critical interpretative synthesis.
- Table 1 - this depends on the level at which you extracted data (quote level or pre-generated theme level), but I wondered if it would be useful to have a column for themes extracted, column for a supporting quote from that study, and a column for the collective interpretation by the author? Perhaps this might all go into a new table? So you would have a study characteristics table and a table of the extracted findings across each study? Then the final table would contain the synthesised findings. It would just tease out the analytic process a little more and help justify your synthesis.
The analysis conducted was very extensive and as such created pages upon pages of analytical processes and codes- too much to submit within the article entry. To provide the reader with further evidence we have created a new Table to provide some example quotations of the identified evidence to support each construct. However we note the length of this table and therefore have referred the reader to the appendices as Table S6
Reviewer 3 Report
This review article provides some valuable insight into numerous barriers to implementation of medical and lifestyle interventions aimed at management of T2D among UK residents of South Asian descent. The Critical Interpretative Synthesis (CIS) approach employed by the authors shows effectiveness in utilizing the data provided in previous qualitative studies to identify how numerous factors, particularly cultural and social, hinder the success of self-management of T2D. The scope of the problem of T2D among SA residents of the UK is significant, therefore this review could provide insight to researchers and clinicians seeking effective strategies to combat T2D.
The technicalities of the methods sometimes become a bit too drawn-out and can sometimes hinder the readability of the article by disrupting the flow of the narrative. Some of the methodological details could be condensed without loss of the essential meaning.
The article really delivers its message beginning in Section 4.2, where the particular interventions (diet, PA, and medical management) are discussed and the specific barriers to positive action are detailed. The logic and evidence for the analysis here is strong and the narrative gathers momentum and makes key points in a readable and intelligible manner. The challenges faced by SA women in the UK are portrayed especially vividly. The cultural, linguistic, and social obstacles to executing self-management behaviors that are generally recognized as effective in mitigating the burden of T2D are comprehensively described. Furthermore, the importance of cultural sensitivity in treating SA patients is made clear by the authors. The use of tables and figures also aids in conveying the main points of the review.
It is recommended to the authors that they perform a thorough proofreading once more to condense the sections containing the methodological details, which is a bit cumbersome to the reader. The quality of the writing is generally very high, with a few notes below:
- Line 85 - plural is "syntheses"
- Line 465 - change to "challenging"
- Line 742-3 phrase beginning with "although..." is not a complete sentence. Revise this section for clarity.
- Line 79 - change to may be from "maybe"
Author Response
Reviewer 3
This review article provides some valuable insight into numerous barriers to implementation of medical and lifestyle interventions aimed at management of T2D among UK residents of South Asian descent. The Critical Interpretative Synthesis (CIS) approach employed by the authors shows effectiveness in utilizing the data provided in previous qualitative studies to identify how numerous factors, particularly cultural and social, hinder the success of self-management of T2D. The scope of the problem of T2D among SA residents of the UK is significant, therefore this review could provide insight to researchers and clinicians seeking effective strategies to combat T2D.
The technicalities of the methods sometimes become a bit too drawn-out and can sometimes hinder the readability of the article by disrupting the flow of the narrative. Some of the methodological details could be condensed without loss of the essential meaning.
The article really delivers its message beginning in Section 4.2, where the particular interventions (diet, PA, and medical management) are discussed and the specific barriers to positive action are detailed. The logic and evidence for the analysis here is strong and the narrative gathers momentum and makes key points in a readable and intelligible manner. The challenges faced by SA women in the UK are portrayed especially vividly. The cultural, linguistic, and social obstacles to executing self-management behaviors that are generally recognized as effective in mitigating the burden of T2D are comprehensively described. Furthermore, the importance of cultural sensitivity in treating SA patients is made clear by the authors. The use of tables and figures also aids in conveying the main points of the review.
It is recommended to the authors that they perform a thorough proofreading once more to condense the sections containing the methodological details, which is a bit cumbersome to the reader. The quality of the writing is generally very high, with a few notes below:
Response:
Thank you for the positive feedback and constructive commentary. We have addressed each of your comments as follows:
We have proofread and in places attempted to condense some aspects of the methodological section e.g. Literature Search and Eligibility paras have been merged.
- Line 85 - plural is "syntheses" amended
- Line 465 - change to "challenging" amended
- Line 742-3 phrase beginning with "although..." is not a complete sentence. Revise this section for clarity. Amended
- Line 79 - change to may be from "maybe" amended